# Diffuse Correlation Spectroscopy: A Review of Recent Advances in Parallelisation and Depth Discrimination Techniques

**DOI:** 10.3390/s23239338

**Published:** 2023-11-22

**Authors:** Edward James, Peter R. T. Munro

**Affiliations:** Department of Medical Physics and Biomedical Engineering, University College London, London WC1E 6BT, UK

**Keywords:** diffuse correlation spectroscopy, cerebral blood flow, laser speckle

## Abstract

Diffuse correlation spectroscopy is a non-invasive optical modality used to measure cerebral blood flow in real time, and it has important potential applications in clinical monitoring and neuroscience. As such, many research groups have recently been investigating methods to improve the signal-to-noise ratio, imaging depth, and spatial resolution of diffuse correlation spectroscopy. Such methods have included multispeckle, long wavelength, interferometric, depth discrimination, time-of-flight resolution, and acousto-optic detection strategies. In this review, we exhaustively appraise this plethora of recent advances, which can be used to assess limitations and guide innovation for future implementations of diffuse correlation spectroscopy that will harness technological improvements in the years to come.

## 1. Introduction

Diffuse correlation spectroscopy (DCS) is a non-invasive optical imaging modality that can be used to measure cerebral blood flow (CBF) in real time [1]. It has important potential applications in clinical monitoring [2] as well as in neuroscience and the development of a noninvasive brain–computer interface [3,4]. However, one of the limitations of DCS is that a trade-off exists between the signal-to-noise ratio (SNR) and imaging depth, and thus brain specificity, of this technique [5]. This is because an increase in imaging depth requires the use of larger source-detector separation (SDS) distances, which result in more photon losses due to absorption and scattering and a subsequent decrease in SNR. An increase in imaging depth also results in the accumulation of phase shifts due to dynamic scattering events, which results in a loss of coherence and SNR. Additionally, as DCS is a diffuse optical technique, it is limited by a lack of inherent depth discrimination within the illuminated region of each source-detector pair, and the CBF signal is therefore also prone to contamination by the extracerebral tissues which the light traverses [6].

DCS was first developed in 1997 [7], and, until the last handful of years, developments in DCS have been fairly quiescent. However, the investigation of novel approaches to improve the sensitivity of DCS to CBF has recently attracted interest from several research groups. Indeed, as was depicted by Carp et al., the field has seen steady linear growth over the past 15 years, with more than 350 publications [8]. Techniques including multispeckle detection strategies [3,9,10,11], time-domain DCS (TD-DCS) [12], DCS in the short-wave infrared region [13,14,15], interferometric approaches [5,16,17,18,19], and acousto-optic modulation [20] have all been proposed. Reducing the cost and enhancing the performance of such systems will allow for more widespread clinical use. There is currently an unmet need to develop such an instrument using a continuous, noninvasive, and portable bedside approach [21], and in this paper, we review the research that addresses this important challenge.

## 2. Motivation

The ability to be able to monitor CBF in a medical setting is of crucial importance [22]. CBF, representing the perfusion of blood through the brain’s tissue beds, needs to be maintained at an average resting value of 40–50 mL per 100 g of brain tissue per minute [23]. This perfusion level is necessary in order to support the brain’s relatively high demand of 20% of the body’s overall oxygen supply [24] as well as to remove the waste products of metabolism. The neurons of the brain require adenosine triphosphate in order to function, the production of which depends almost exclusively on oxidative processes [25]. Inadequate CBF, even if only for a few minutes, can lead to irreversible brain damage, ischaemic stroke, and death [22]. Excessive CBF can cause damage to the blood–brain barrier, which can cause seizures, headaches, encephalopathy, and stroke [26]. Durduran and Yodh therefore state that [24],

‘CBF is an important biomarker of brain health and function. It is a critical physiological parameter linking metabolic demand to oxygen supply, oxygen supply to metabolic consumption, and metabolic consumption to byproduct clearance.’

CBF monitoring is critical in unconscious or anaesthetised patients as it provides an indicator of the adequacy of the delivery of vital nutrients to the brain as well as a commentary on the efficacy of cerebral autoregulation (CA). CA is the intrinsic homeostatic mechanism by which CBF is kept within tightly controlled bounds in the face of variations in mean arterial pressure (MAP) and intracranial pressure (ICP) [23], which itself can affect arterial and venous pressure changes within the skull.

Cerebral perfusion pressure (CPP) is the net pressure gradient that drives CBF and causes the tissues of the brain to be perfused with blood. CPP can be calculated as the difference between MAP and ICP [26,27]: (1)CPP=MAP−ICP.

MAP is also known as arterial blood pressure (ABP) or systemic arterial pressure (SAP). MAP in the brain is similar to the MAP anywhere else in the human body and can be determined by intraarterial catheterisation. MAP can also be approximated noninvasively by
(2)$$\mathrm{MAP}=\mathrm{DP}+\frac{\mathrm{SP}-\mathrm{DP}}{3},$$
where SP is systolic pressure and DP is diastolic pressure.

ICP is the pressure within the rigid human skull, the volume of which, according to the Monro and Kellie doctrine, cannot change [28]. ICP is the sum of the partial pressures of brain tissue, cerebrospinal fluid (CSF), and cerebral blood volume (CBV) contained within the skull and thus may vary with any of these three partial pressures [23]. Changes in ICP can occur in the healthy patient due to arterial pulsations within the brain, changes in head and body positioning, or exercise, but may also occur due to the administration of drugs or various disease states, such as traumatic brain injury, intracranial haemorrhage or tumour, disorders of CSF circulation, and stroke [26,28].

Analogous to Ohm’s or Darcy’s Law, CBF varies with CPP according to [26]: (3)$$\mathrm{CBF}=\frac{\mathrm{CPP}}{\mathrm{CVR}},$$
where CVR is total cerebrovascular resistance and is associated with the entirety of the brain’s vascular tree. Thus, one component of CA involves *pressure* autoregulation, by which varying degrees of vasoconstriction and vasodilation are used to affect changes in CVR. The other component of CA is *metabolic* autoregulation, in which CBF is adjusted according to the metabolic needs of the tissue. This metabolic effect is primarily attributed to changes in brain tissue pH [23].

In order to assess CA in a continuous or semi-continuous manner in hospitalised patients, invasive measurements of ICP may be taken in order to calculate an estimate of CPP, which may be used as a surrogate for CA assessment [27]. For example, in the absence of CA, an increase in MAP will cause an increase in CPP [26]; alternatively, a decreased CPP can be viewed as an indicator of low CBF and impaired CA. This also allows clinicians to manage ICP directly but has two disadvantages:in certain populations of patients, clinicians would prefer to avoid the risks of invasive ICP measurement using continuous cerebral monitoring devices [29], such as the risk of bleeding, infection, and misplacement [30];a more accurate assessment of CA may be inferred from direct measurements of local CBF [24], and evaluation of its correlation with MAP, together with evaluation of clinical neurological signs. There should be no correlation between CBF and MAP when CA is operating effectively.

Indeed, raised ICP and low CBF are associated with ischaemia and poor outcomes following brain injury, and so many management protocols target these two parameters directly [28] (N.B. a recent review of non-invasive ICP monitoring methods is available in [31]).

CBF measurements may also be used to distinguish between hypoxia and ischaemia, to avoid hyperperfusion injury, to characterise different types of hypoxic states, as well as to examine connections between vascular physiology and neurophysiology (i.e., neuro-vascular coupling) in healthy patients for neuroscience applications [25]. Finally, CBF monitoring also has applications in the development of a noninvasive brain–computer interface [3,4], although these authors have demonstrated that sensitivity to local, subsecond latency haemodynamic response of the cerebral cortex at a depth of 15 mm below the scalp surface is not feasible with current detector technology and signal processing techniques.

Many imaging modalities exist that attempt to measure CBF, each of which has its own distinct advantages and disadvantages (the interested reader is referred to [26,32] for comprehensive reviews of non-optical and optical CBF measurement methods). Within the more general context of tissue blood flow (BF) measurement, Yu et al. [21] state,

‘The ideal BF measurement should provide quantitative information about macro- and microvasculature with millisecond temporal resolution. The measurements should be carried out continuously, noninvasively, and without risk to subjects. Furthermore, ideal measurements would not be limited to the tissue surface, i.e., it is desirable to probe BF in deep tissues. Unfortunately, no such ideal modality exists’.

The motivation to measure local microvascular BF is by no means limited to the brain. Abnormal BF can be seen in a variety of medical conditions in the various tissues and organs of the human body, and the utility of its measurement has been demonstrated in conditions such as cancer and peripheral arterial disease (PAD), as well as in the monitoring of muscle disease, and normal exercise physiology. A few of these applications are briefly outlined below.

Worldwide, breast cancer is the most frequently diagnosed and primary cause of cancer-related mortality amongst women [33]. Therefore, every incremental improvement in breast cancer detection could have a significant impact on the detection of this disease. Previous studies have shown that BF in tissues affected by breast cancer is larger than in normal tissues [34,35]. BF measurements have been used to monitor the response of:Breast cancer to chemotherapy [35];Head-and-neck cancer to chemoradiation therapy [36];Prostate cancer to photodynamic therapy (PDT) [37].

It has also been demonstrated that BF measurement can be used in the prediction of the response of murine tumours to PDT [38] as well as in the real-time in situ monitoring of the response of prostate cancers to PDT [37].

Measurement of BF in skeletal muscles has important applications in exercise medicine and the furtherance of our knowledge of exercise physiology, as well as in the aiding of our understanding of diseases such as PAD, and how muscle function is affected by cardiovascular disease more generally [39]. There has also been recent interest in BF imaging of the heart to assess cardiac tissue for evidence of myocardial infarction (i.e., heart attack) [40], which would be especially useful in the emergency setting.

DCS is the archetypal optical modality to measure flow beyond a few millimetres of the tissue surface, and it is suitable for continuous, noninvasive, portable, and real-time CBF measurement. However, the depth penetration and spatial resolution of DCS are fundamentally limited by the nature of diffuse optics, and, in this paper, we review the various approaches that have been employed to overcome these limitations.

## 3. Laser Speckle

Laser speckle is a random interference pattern that is produced by the coherent summation of a scattered laser beam of light, each component of which has travelled a slightly different pathlength due to scattering. Movement of scattering red blood cells (RBCs) in biological tissue causes modifications to the intensity of this speckle pattern, in both the temporal and spatial domains [41]. Spatial ensemble methods use a high-pixel-count and typically low-frame rate detector to capture speckle patterns and infer sample dynamics through assessment of the spatial fluctuations of speckle [42].

Alternatively, temporal sampling methods record the intensity fluctuations of each pixel over time, typically using high-frame rate and single or few-mode detection [42]. When an incident coherent light beam interacts with a scattering particle, each particle develops an induced dipole moment. Each of these oscillating dipoles emits scattered light fields in all directions. The scattered light electric field at the detector, E(t), is thus composed of the addition of all of these oscillating dipole contributions. Due to the movement of the scattering particles, the phases of these fields vary relative to each other, which results in fluctuations in the field, and therefore intensity, over time. Temporal sampling methods access sample dynamics by considering the autocorrelation of such a time series of measurements, the rate and shape of the decay of which correspond to the speed and the nature of scattering particles, respectively.

In practice, the scattered normalised intensity temporal autocorrelation function, $g_{2}\left(\tau \right)$, at the detector is measured. This is more straightforward to measure than the electric field [24], which fluctuates at 0.23–0.46 PHz in the near-infrared (NIR) optical window (650–1300 nm) and which light sensors are unable to detect. However, $g_{2}\left(\tau \right)$ is directly measurable as [21]
(4)$$g_{2}\left(\tau \right)\equiv \frac{\langle I(t)I(t+\tau )\rangle }{{\langle I\left(t\right)\rangle }^{2}},$$
where intensity $I\left(t\right)={\left|E\left(t\right)\right|}^{2}$, τ is the autocorrelation lag time, and 〈⋯〉 denotes a spatial ensemble-average (which is equivalent to a time-average for an ergodic sample [7]). An example of $g_{2}\left(\tau \right)$ is shown in Figure 1. The minimum lag time (or equivalently the maximum frame rate of detection) is an important property in DCS measurements, as it determines how accurately $g_{2}\left(\tau \right)$ can be measured within the timescales of interest. In order to characterise $g_{2}\left(\tau \right)$ for in vivo experiments, a minimum lag time of 1 μs is typically required [29], and it has also been demonstrated that earlier lag times are more sensitive to deeper photon paths [43], which is a consideration when discriminating the depth within the sample from which the measured signal has originated, as described in Section 4.5. Assuming that the electric field is a zero-mean Gaussian random variable, an assumption that breaks down when scattering sites are few or correlated, the Siegert relation, can then be used to extract the electric field autocorrelation function, $g_{1}\left(\tau \right)$, from $g_{2}\left(\tau \right)$ using [44,45]
(5)$$g_{2}\left(\tau \right)=1+\beta {\left|g_{1}\left(\tau \right)\right|}^{2},$$
where [7]
(6)$$g_{1}\left(\tau \right)=\alpha \left|g_{1d}\left(\tau \right)\right|+\left(1-\alpha \right),$$
where $g_{1}\left(\tau \right)$ is the field autocorrelation due to *both* dynamic and static scatterers, $g_{1d}\left(\tau \right)$ is the field autocorrelation due to dynamic scatterers *only*, and where α∈[0,1] and β∈[0,1] are both unitless factors.

Sample motion may be inferred by fitting measured data to a theoretical model of $g_{1}\left(\tau \right)$ that incorporates mean-square particle displacement, $\langle \Delta r^{2}\left(\tau \right)\rangle$ (which directly characterises particle displacement). For the case of deterministic convective motion of scatterers, $\langle \Delta r^{2}\left(\tau \right)\rangle =\langle V^{2}\rangle \tau^{2}$, where $\langle V^{2}\rangle$ (cm^2^/s^2^) is the second moment of the speed distribution of scattering particles. In the case of diffusive Brownian motion of scatterers $\langle \Delta r^{2}\left(\tau \right)\rangle =6D_{b}\tau$, where $D_{b}$ (cm^2^/s) is the effective Brownian diffusion coefficient of scattering particles [46]. Other types of particle motion contribute toward the measured signal (e.g., rotation, shear flow, and turbulence) but the literature focuses on these two effects. Diffusing light is primarily absorbed when crossing large arteries and veins, and therefore DCS is most sensitive to the motion of scatterers in the *microvasculature* (capillaries, arterioles, and venules) rather than macrovasculature [29]. Microvasculature is convoluted and the distribution of the direction of velocities of RBCs can therefore be assumed to be isotropic. Previous authors have found that the Brownian model fits observed data more precisely than the convective flow model in many biological tissues [21], although a mixed model can also be used [47], as shown by Equation (11).

α is an adaptation to biological tissue and refers to the fraction of scattering events due to dynamic, rather than stationary, scatterers. This factor is therefore the ratio of dynamic scatterers to the total number of scatterers in a sample. One must be careful when applying the Siegert relation to samples that have a large proportion of static scatterers; however, Durduran et al. concluded that ‘one can routinely employ the Siegert relation in most tissue dynamics experiments, except perhaps those wherein the subject is exercising’ [29]. If we make the assumption that a sample is composed entirely of dynamic scatterers, then Equation (6) can be simplified to
(7)$$g_{1}\left(\tau \right)=\left|g_{1d}\left(\tau \right)\right|.$$

Similarly, if a sample is composed entirely of static scatterers, then we have
(8)$$g_{1}\left(\tau \right)=1.$$

β, the coherence factor, is a constant determined by the geometry and collection optics of the experiment and is equal to 1 ideally. β is inversely proportional to the number of detected speckles, and is also related to the coherence length, stability of the laser light source, stray light, detector stability, sample coupling, and polarisation state [7]; it can be determined as the value of $g_{2}\left(0\right)$.

## 4. Diffuse Correlation Spectroscopy

DCS typically employs a long coherence length (∼10 m) laser light source, which is mounted on the sample surface and delivered through a multimode source fibre. A long coherence length is used to provide the necessary conditions for interference over the diffuse optical pathway between the source and the detector. Surface mounted single-mode or few-mode optical fibres are then used to collect photons from a single/few speckles, using SDS distances of 2.5–3.5 cm. Fast photon-counting avalanche photodiodes (APDs) can be used as detectors, as can photon multiplying tubes (PMTs) or single photon avalanche detectors (SPADs). A correlator board receives the intensity output from the photodetector and then computes $g_{2}\left(\tau \right)$, typically using the multi-tau autocorrelation algorithm [21]. Briefly, and as described more fully in the remainder of this section, the shape and rate of decay of $g_{2}\left(\tau \right)$ provides information corresponding to the nature and the speed of the motion of scatterers in the illuminated volume (depicted by the red banana-shaped region in Figure 1). A typical DCS system requires only one wavelength of NIR light to operate (commonly ∼785 nm), and can be made to be very compact due to the advent of solid state laser technology. A schematic of the entire DCS measurement process is shown in Figure 1.

Boas and Yodh derived the correlation diffusion equation [7], the theoretical basis of DCS, which describes the propagation of the temporal electric field autocorrelation function in turbid biological tissue, and provides a framework for the study of tissue dynamics and tomography [21]. (It should be noted that DCS is a differential formulation of diffusing wave spectroscopy (DWS), which models the autocorrelation function as an integral over photon pathlengths [48].) The diffusion equation for electric field correlation (i.e., the correlation diffusion equation) can be derived for a clinically relevant semi-infinite geometry (Figure 2) [1], in which the *unnormalised* solution for the autocorrelation function for dynamic scatterers is given by [7].
(9)$$G_{1d}\left(\tau \right)=\frac{S_{0}}{4\pi D}\left[\frac{\exp^{-K\left(\tau \right)r_{1}}}{r_{1}}-\frac{\exp^{-K\left(\tau \right)r_{2}}}{r_{2}}\right].$$

The *normalised* temporal electric field autocorrelation function is then [35]
(10)$$g_{1d}\left(\tau \right)=\frac{G_{1d}\left(\tau \right)}{G_{1d}\left(\tau =0\right)},$$
where, in Equation (9),

$S_{0}$ is the optical source intensity;*D* is the optical diffusion coefficient, $\frac{1}{3(\mu_{a}+\mu_{s}^{\prime })}$, where $\mu_{a}$ is the absorption coefficient and $\mu_{s}^{\prime }$ is the reduced scattering coefficient;$K\left(\tau \right)=\sqrt{3\mu_{a}\mu_{s}^{\prime }+\mu_{s}^{\prime 2}k_{0}^{2}\langle \Delta r^{2}\left(\tau \right)\rangle }$ is the decay constant, where $k_{0}=2\pi n/\lambda$ is the wavenumber of the incident light field, and *n* is the refractive index of tissue;$z_{0}=1/\mu_{s}^{\prime }$ is the depth into the medium at which the collimated source is approximated as a positive isotropic source;ρ is the distance between the optical source and detector;$R_{\mathrm{eff}}=-1.440n^{-2}+0.710n^{-1}+0.668+0.0636n$ is the effective reflection coefficient and accounts for the reflective index mismatch between air ($n_{\mathrm{out}}$) and tissue ($n_{\mathrm{in}}$), where $n=n_{\mathrm{in}}$/$n_{\mathrm{out}}$;$z_{b}=\frac{2z_{0}}{3}\frac{\left(1+R_{\mathrm{eff}}\right)}{\left(1-R_{\mathrm{eff}}\right)}$, $-z_{b}$ is the position at which there should be a signal size of zero to fulfil the extrapolated boundary condition [49];$r_{1}=\sqrt{z_{0}^{2}+\rho^{2}}$ is the distance between the detector and an approximated positive isotropic imaging source;$r_{2}=\sqrt{{\left(2z_{b}+z_{0}\right)}^{2}+\rho^{2}}$ is the distance between the detector and an approximated negative isotropic imaging source located at position $z=-(z_{0}+2z_{b})$.

A value for $\langle \Delta r^{2}\left(\tau \right)\rangle$ can be extracted from measured data by fitting to either a Brownian or a convective motion model of Equation (10) or a linear combination of the two, assuming that RBCs are the only source of dynamic scattering events and that [47]
(11)$$\langle \Delta r^{2}\left(\tau \right)\rangle =6D_{b}\tau +\langle \Delta V^{2}\left(\tau \right)\rangle \tau^{2}.$$

The interested reader is referred to [50] for examples of fitting $D_{b}$ and $\langle V^{2}\rangle$ to experimental data and also for examples of mixed model fitting and the validation of measured $D_{b}$ values in an intralipid phantom.

Figure 2 describes the method of images and the notation that is used in a semi-infinite DCS geometry model when ensuring an extrapolated-zero boundary condition.

$D_{b}$ can be used as a blood flow index (BFI) parameter, although it has units of cm^2^/s, rather than the more commonly encountered blood perfusion unit of ml/100g/min. $D_{b}$ is also a relative, rather than an absolute, measure. Relative change in BF, $r\mathrm{BF}=\mathrm{BFI}/{\mathrm{BFI}}_{0}$, where BFI_0_ is the baseline measurement of BFI, acquired by DCS measurement has been shown to agree with relative changes in absolute BF as measured by gold standard techniques, such as arterial spin labelling magnetic resonance imaging (ASL-MRI) [25]. Additionally, the use of hybrid DCS/near-infrared spectroscopy (NIRS) systems to measure NIRS derived tissue oxygen saturation and DCS derived CBF can be utilised to provide quantitative measurement of cerebral metabolic rate of oxygen extraction (CMRO_2_), and its continuous noninvasive inference in this context relies on fewer assumptions than when using NIRS alone [29]. Local CMRO_2_ is an important physiological parameter to monitor as it is a function of oxygen saturation of arterial and venous ends of the local cerebral circulation as well as CBF itself. Overall, this provides a more robust picture of brain health [25]. Validation studies show that DCS measurements of BF are in close agreement with results obtained by theoretical expectation, computational simulation, and other biomedical imaging modalities [21].

### 4.1. Key Limitations

The conventional implementations of DCS that have been described in the literature typically employ single-mode photon counting techniques, with an associated high cost of detection components. Such methods are limited by low light throughput [25] in a single-mode, placing a lower limit on the detection time [51]. Increasing penetration depth requires the use of larger SDS distances, which will decrease the available SNR further, by increasing the number of absorption and scattering events, since the attenuation of NIR light by these two mechanisms is in the order of 10 dB/cm [52], and this can be worsened by the presence of hair or darkly pigmented skin.

Increasing acquisition time can ameliorate this situation but leads to a reduction in temporal resolution. Taking the average of many single-mode detection fibres bundled together is an expensive option that requires many photon counting detectors and can increase the complexity of system integration. Improved collection optics, the use of few-mode detection fibres, and increasing the amount of light delivered to the tissue can also help to improve the SNR [25,53,54]. However, patient safety limits [55] must be adhered to, which necessitates an optical source of sufficiently large diameter and low power rating. All the above characteristics limit the applicability of conventional homodyne DCS in portable continuous monitoring applications to which optical methods are otherwise well suited.

Light collection efficiency and SNR increase with the effective number of detected modes, *N*. The SNR also increases with β, but this quantity scales as 1/N for each detected mode due to loss of coherence. In typical non-ideal situations, the SNR may be optimised by using a single-mode detector fibre; however, if the optimisation of temporal resolution is of interest, then a few-mode detector fibre can be used [56]. The use of a single-mode fibre in a conventional DCS experiment will lead to a maximum β value of 0.5, due to the two orthogonal electric field components collected by the fibre. The recorded value of β will likely be less than this due to variations in light coherence, laser stability, stray light, detector stability, and fibre-tissue coupling [46]. This value of β can be doubled by placing a polariser in front of the detector fibre.

DCS offers the ability to perform safe, noninvasive, and continuous bedside measurements of superficial microvascular CBF, with a relatively high temporal resolution (up to 100 Hz) [21]. However, it suffers from poor spatial resolution (which worsens with depth) and depth penetration (a general rule of thumb in diffuse optics is that a banana-shaped region of depth equal to one-half to one-third of the optical SDS distance is sampled [25,29]). The main limitation of DCS is that a trade-off exists between the SNR and imaging depth, and thus brain specificity, of this technique [5]. Additionally, DCS is also susceptible to superficial extra-cerebral signal contamination, and the measurement of absolute CBF with this technique still remains challenging [26]. DCS also depends on the accurate measurement of $\mu_{a}$ and $\mu_{s}^{\prime }$ to provide a precise assessment of CBF [57], and it is also limited by low SNR due to single-mode detection (especially when using large SDS values, in the presence of hair, and in patients with darkly pigmented skin) and susceptibility to motion artefacts.

Despite these disadvantages, DCS has important potential applications in clinical monitoring [2,15], as well as in neuroscience and the development of a noninvasive brain-computer interface [3,4], as introduced in Section 1. Previous authors have noted that improvements to increase SNR, depth penetration, and both spatial and temporal resolution will assist the development of DCS functional experiments as well as expand the range of uses for DCS in clinical monitoring [25]. The utility of high-frame rate (∼20 Hz) DCS measurements, compared to the frame rates of 0.3 to 1 Hz that have been traditionally used, has also been discussed [1]. These benefits include improved monitoring of cerebrovascular autoregulation dynamics, more robust identification of motion artefacts, and increased throughput that could enable high spatial resolution with fewer detectors.

As introduced to the reader in Section 1, the investigation of novel approaches to improve the sensitivity of DCS to CBF has therefore recently attracted interest from several research groups. The remainder of Section 4 and the entirety of Section 5 are accordingly dedicated to the review of the wide variety of these approaches that have recently been published.

### 4.2. Multispeckle Approaches

As described in Section 4.1, it has generally been accepted in the field of DCS that as the number of modes per detection fibre increases, the resulting gain in SNR due to increased photon intensity is negated by the subsequent loss of coherence and reduction in β. However, closer inspection of the DCS noise model (Equation (8) of [58]) reveals one term in which the photon count has a higher weighting than β, which can be made use of in low photon count environments. Carp therefore proposed that the use of few-mode detection, rather than single-mode detection, could confer a slight SNR advantage at low photon count rates and demonstrated a 20% reduction in the standard deviation of relative BFI when switching from single-mode to few-mode detection, when making DCS measurements on an intralipid phantom with an SDS distance of 3 cm [59].

However, this SNR benefit that results from integrating multiple mutually incoherent speckle grains on the same photodetector does not persist beyond few-mode detection, and DCS therefore clearly benefits from a parallelised detection strategy, in which the detection of *N* speckle grains on multiple independent photodetectors will result in a $\sqrt{N}$ improvement in the SNR, under the assumption of shot noise limited detection [10]. Dietsche et al. validated this approach by bundling together 28 stand-alone SPADs, which yielded a $1/\sqrt{28}$ reduction in noise [53]. Johansson et al. demonstrated a more compact approach by developing a 5×5 SPAD array for DCS detection, which resulted in improved SNR [60].

Subsequently, a 32×32 SPAD array (with a component cost of GBP 34,000), was used for DCS detection by two groups [3,10], which allowed Sie et al. to demonstrate a 32-fold increase in SNR with respect to traditional single speckle DCS, for $g_{2}\left(\tau =4\;\mu s\right)$ [3]. More recently, the same group published an SNR gain of ∼470 over single speckle DCS, using a 500×500 SPAD array at 785 nm [11], albeit with a photon detection efficiency (PDE) of ∼15% (conventional single SPADs have a PDE of ∼70% at this wavelength). The authors were able to achieve a frame rate of 0.09 MHz when using all available pixels but were able to increase this to 2.63 MHz when using 3.2% of all available pixels, and as such, there is a tradeoff between pixel count and minimum lag time. Given this, it is of note that DCS relies on the detection of optical fluctuations in the 1 MHz regime (especially for deep tissue measurements [5,17]), and that acousto-optic modulated DCS (AOM-DCS), which is described in Section 4.7, is typically employed using ultrasound frequencies of 1–5 MHz [51] (conventional single SPADs have a frame rate of ∼10 MHz). Furthermore, Wayne et al. did not describe what the noise floor of their instrument is, nor how the noise floor does not become a limiting factor with regard to SNR gain when averaging over 250,000 pixels. An understanding of this is vital in order to analyse the application of even larger SPAD arrays to DCS, which will surely be explored in the years to come [4]. Apart from the high component cost, another factor to consider when using SPAD arrays is the extremely high data rate when independently autocorrelating multiple photon arrival time streams in parallel. Della Rocca et al. were able to calculate 12,288 autocorrelation functions in real time from the output of half of a 192×128 SPAD array by using a FPGA-embedded autocorrelation algorithm, which yielded a 110-fold increase in SNR with respect to traditional single speckle DCS [61]. Finally, a 32×32 InP/InGaAs SPAD array for long wavelength TD-DCS is currently being developed [62], and this is described in Section 4.6. Intense research effort is currently underway to improve the performance of SPAD array technology in terms of increased pixel count, shorter exposure time, higher PDE (especially at longer wavelengths), and high-performance data processing [61].

A third approach to multispeckle detection is to use conventional camera array detectors; however, current camera technology is limited by noise performance (when compared to SPADs) and by detection times, which are not short enough to fully characterise $g_{2}$ at very short lag times. By making use of the relationship [63,64]
(12)$$K=\frac{\sigma }{\mu }={\left[\frac{2\beta }{T}\int_{0}^{T}\left(1-\frac{\tau }{T}\right){\left[g_{1}\left(\tau \right)\right]}^{2}\;d\tau \right]}^{1/2},$$
where *K* is speckle contrast, σ and μ are the standard deviation and mean of a spatial ensemble of speckles, respectively, and *T* is camera exposure time, Murali et al. demonstrated that by making measurements of *K* using spatial sampling for various values of *T*, it is possible to recover $g_{1}\left(\tau \right)$ using low-cost and low-frame rate cameras [9,65,66]. It is of note that Xu et al. found that spatial sampling methods outperform temporal sampling methods in terms of SNR in the case of optical brain monitoring due to the interplay of the limited photon flux and the number of independent observable (NIO) speckles involved [42]. This is because the limited time measurement duration necessitated by in vivo monitoring constrains NIO speckles for temporal sampling but not for spatial sampling.

With regard to compensating for camera noise performance, interferometric approaches have been proposed, and these are discussed in Section 4.4 and Section 5. More recently, Robinson et al. demonstrated a multispeckle, long wavelength, interferometric DCS (LW-iDCS) system [15], and this is more fully expounded in Section 4.4 also. Care must be taken when using multimode fibres for detection, as these fibres are sensitive to motion caused by environmental vibrations, for example [15,59].

### 4.3. Long-Wavelength Approaches

Compared to a more commonly used wavelength of 785 nm, using a design wavelength of 1064 nm confers an SNR advantage in DCS systems. Light suffers from fewer optical scattering events at this wavelength, where a local minimum in water absorption combined with lower haemoglobin absorption also occurs, which results in an increase in photon availability [67]. Maximum permissible exposure limits are also higher at 1064 nm, meaning that roughly 2.5–4 times more optical power can be used [59]. The decrease in scattering also results in a slower decay of the autocorrelation curve, which moves $g_{1}$ into longer and less noisy, and thus more easily detectable, timescales. By considering this overall photon budget, Carp et al. demonstrated that at 1064 nm, 13, 10.5, and 7 times more photons will be detected than at 765, 785, and 850 nm, respectively [13]. These authors note that the reduction in scattering that occurs at 1064 nm results in a *reduced* sensitivity to motion; however, this is more than compensated for by the advantageous photon budget at this wavelength. It is of note that the low haemoglobin absorption that occurs at 1064 nm makes this choice of wavelength impractical for NIRS experiments.

The main limitation that currently prevents the implementation of DCS at 1064 nm is a lack of suitable detector technology. For example, silicon SPADs have a PDE of 64% and 54% at 765 and 850 nm, respectively, but which drops to 3% at 1064 nm [13]. Alternatively, InGaAs SPADs have a PDE of ∼32% at 1064 nm; however, these detectors have unacceptably long afterpulsing in the region where $g_{2}$ starts to decay (i.e., 1–10 μs) [14]. Potential solutions to this afterpulsing problem include custom detector designs, and cross-correlation approaches using two detectors [13,17,68]. More recently, LW-iDCS has been proposed (Section 4.4), which makes use of multispeckle interferometric detection by a fast InGaAs linescan camera to mitigate the negative aspects of detector technologies when working at 1064 nm [15]. A second detector technology that is suitable for 1064 nm is superconducting nanowire single-photon detection (SNSPD). SNSPD technology possesses several advantages over InGaAs SPADs, including shorter dead time (<50 ns), better timing resolution (<80 ps), an extremely favourable dark count rate of 1 count per second (CPS), a PDE of >80% at 1064 nm, and no afterpulsing issues. Ozana et al. achieved an SNR gain of 16, for $g_{2}\left(\tau =4\;\mu s\right)$, when making in vivo measurements on the forehead of 11 human subjects with an SDS distance of 25 mm, compared to DCS at 850 nm using silicon SPAD detection [14]. However, SNSPD technology is expensive, bulky, and loud, requires cryostats to achieve an operating temperature of 2–3.1 K, and has a turn-on time of several hours, which limits its clinical application at present. Additionally, although SNPSD units with 16 channels or more are beginning to be offered [69], the use of SNSPD does not lend itself well to high channel count systems.

Long-wavelength, multichannel TD-DCS systems are currently being developed ([62], for example), and these are described further in Section 4.6.

### 4.4. Interferometric Approaches

The theory of DCS that has been considered so far has been restricted to homodyne detection (i.e., the self-interference of changing light fields) in which $g_{2}\left(\tau \right)$ is related to the motion of the sample by the Siegert relation (Equation (5)). An interferometric DCS (iDCS) system may be constructed by modifying a homodyne DCS setup with a pair of fibre couplers to create a Mach–Zehnder interferometer. iDCS works by using the reference arm of the interferometer to coherently amplify the speckle fluctuations of the sample arm, and the expression for $g_{2}\left(\tau \right)$ can then be found as [17]
(13)$$g_{2}\left(\tau \right)=1+\beta {\left(1-\frac{I_{R}}{I_{T}}\right)}^{2}{\left[g_{1}\left(\tau \right)\right]}^{2}+2\beta \frac{I_{R}}{I_{T}}\left(1-\frac{I_{R}}{I_{T}}\right)g_{1}\left(\tau \right),$$
which can be simplified to
(14)$$g_{2}\left(\tau \right)=1+\beta_{1}{\left[g_{1}\left(\tau \right)\right]}^{2}+\beta_{2}g_{1}\left(\tau \right),$$
where $I_{R}$ is the reference intensity, and $I_{R}/I_{T}$ is the fractional reference intensity, where $I_{T}=I_{s}+I_{R}$ is the total intensity, and $I_{S}$ is the sample arm intensity. A caveat of this technique is that any laser intensity instability will have a larger influence on $g_{2}\left(\tau \right)$ for an interferometric measurement [59].

Interferometric techniques are beneficial in that they can compensate for detector nonidealities (e.g., afterpulsing, read noise, and dark noise) [17], they are robust to environmental noise (such as ambient light that is present in the clinical environment) [59], and they allow for effective measurement even in the presence of very low signal levels, allowing for the measurement of CBF within short acquisition times in low-light conditions [19]. This means that less expensive detectors with sub-optimal noise characteristics can be used for interferometric detection.

Using silicon SPADs and a 785 nm laser, Robinson et al. used iDCS to demonstrate an improvement in the SNR of $g_{1}\left(\tau \right)$ by approximately a factor of 2 at each time lag [17]. These authors also showed an up to 80% reduction in the variability of measured BFI using this technique. The same group later presented LW-iDCS, as introduced in Section 4.2 and Section 4.3, which makes use of multispeckle, interferometric detection at 1064 nm using a fast InGaAs linescan camera operating at 300 khz [15]. This technique was used to demonstrate a 4.5-fold improvement in SNR over homodyne SNSPD long wavelength DCS (LW-DCS) for high-speed pulsatile flow measurement for a single-channel comparison. Furthermore, compared to homodyne SNSPD LW-DCS, LW-iDCS is a cart-based system and also offers a ∼7-fold cost reduction.

Another group presented interferometric DWS (iDWS): an approach which makes use of multimode fibre detection and a high speed linescan CMOS camera capable of operating at 333 kHz [16]. The use of camera-based detection is conducive to multispeckle detection and the SNR advantage that this confers, and this system is capable of measuring ∼96 speckles simultaneously, meaning that clear BFI traces can be obtained using an SDS distance of 35 mm on the adult human head with an integration time of 0.1 s [6]. Although LW-iDCS and iDWS are robust to the effects of ambient light, their sample rates are considerably slower than a conventional DCS/DWS system (which is typically in the order of ∼10 MHz) and they cannot resolve sample decorrelations shorter than 3 μs. Additionally, these approaches do not have the temporal resolution necessary to sufficiently resolve ultrasound tagged photons (Section 4.7), which are typically modulated by ultrasound pressure fields fluctuating in the range of 1–5 MHz [51].

The authors of the iDWS technique have also recently proposed multi-exposure iDWS (MiDWS), which makes use of a low-frame rate 2D CMOS sensor and which measures brain BFI using an SDS of 3 cm and an integration time of 0.096 s at a lower cost per pixel than iDWS [5].

Finally, Zhao et al. recently published iDWS with an electronically variable time of flight (TOF) filter [70,71], a technique that is able to measure 200 autocorrelation functions in parallel with TOF discrimination. This paper represents a step change in interferometric diffuse optics, being the first demonstration of a scalable approach to acquiring TOF discriminated autocorrelation functions. The technique works by rapidly tuning the wavelength of a narrowband laser during the exposure time of a linescan CMOS detector, thereby reducing the coherence length of the source and creating an adaptable and variable TOF filter. Using an SDS distance of 1 cm on the forehead of human subjects and a deep TOF filter, the authors were able to achieve a 2.7-fold reduction in scalp sensitivity compared to CW-DCS. Similar to TD-DCS (Section 4.6), the short SDS distances that this technique employs afford a subsequent improvement in spatial resolution.

BFI measurement using holographic approaches has also been documented by two groups, and these approaches are enumerated here. As digital holography can be a lensless technique, it may be free from aberrations by imaging devices [72]. Digital holography is inherently interferometric, and the use of area sensors also allows for multispeckle detection as well as facilitating the implementation of tomographic and depth discrimination techniques. Above in this section, and in Section 4.2, we have outlined that interferometric and multispeckle detection are approaches that have recently been investigated to improve the SNR and depth specificity of DCS. Both of these approaches can be combined into one detection modality in a digital holography system. Additionally, shot noise limited detection can be achieved in digital holography by combining an appropriate temporal filtering strategy with the spatial filtering that is facilitated by off-axis interference of sample and reference arms [73,74], which can lead to further SNR gains. Digital holography can also be used to implement long wavelength approaches, as NIR-enhanced silicon cameras and InGaAs cameras can be employed for detection. Speckle detection systems must use a coherent light source, and therefore the requirement for coherence that digital holography presents does not represent an additional hardware cost for speckle detection. Finally, when compared to SPADs and SNSPD technology, cameras offer an extremely low relative cost of detection.

Digital holography is therefore an attractive modality with which to detect speckle patterns in biomedical optics; indeed, it has been highlighted as a promising detection strategy for acousto-optic tomography (AOT) [75]. By shifting the frequency of the reference arm near to the acoustic sideband, it is possible to selectively detect tagged photons at the shot noise limit using off-axis holography [76,77]. Most recently, Hussain et al. developed an analytical model for detection of an AOT signal using digital holography with a frequency shifted reference arm, which they implemented to successfully resolve flow in a dynamic flow phantom [78]. These authors showed that the AOT signal within the ultrasound focus was dependent on both the flow through that volume and the integration time of the camera and were able to achieve in vitro depth-resolved flow measurements by analysing images captured using different integration times.

Fourier domain DCS (FD-DCS) [50] and interferometric speckle visibility spectroscopy (ISVS) [19] are two techniques that make use of off-axis holographic camera-based detection of multispeckle patterns to determine microvascular BF. Briefly, FD-DCS employs a Mach–Zehnder interferometer where light from the sample arm interferes with frequency shifted light from the reference arm. Detecting the result of interference between the sample and the reference arms, for different reference light detuning frequencies, ω, removes the need to detect very rapid intensity changes when frequency shifting is not used, as is required in conventional DCS experiments. This allows for a slower detector to be used, such as a relatively inexpensive camera. Thus, FD-DCS, which is inherently an interferometric technique, also lends itself well to multispeckle detection. Additionally, the interferometric measurement interrogates the electric field directly, rather than intensity, and therefore the Siegert relation, and the assumptions therein, do not constrain FD-DCS [19]. According to the Wiener–Khinchin theorem, the first-order power spectral density (PSD) of the field fluctuations due to dynamic scatterers, $s_{1d}\left(\omega \right)$, is the Fourier transform of the field autocorrelation function, $g_{1d}\left(\tau \right)$ [79,80,81,82]: (15)$$s_{1d}\left(\omega \right)=\int_{-\infty }^{+\infty }g_{1d}\left(\tau \right)\exp^{-i\omega \tau }\;d\tau ,$$
and thus an FD-DCS measurement and a conventional DCS measurement contain entirely equivalent information [7]. FD-DCS was used to measure microvascular BF at a frame rate of 10.8 Hz in the human forearm using an SDS distance of 1.13 cm [50], and further signal processing improvements yielded an SNR gain of 36 whilst detecting up to ∼1290 speckles in parallel (the largest SNR gain in the DCS literature at the time, cf. Section 4.2) [83].

A fundamental limitation of FD-DCS to the application of in vivo measurement is the requirement to obtain a series of very short exposure camera frames, at a series of discrete frequency shifts, all at a fast enough sample rate so as to ensure the accurate recovery of pulsatile information. These short camera exposure times come at the cost of broadening of the signal in the Fourier domain, which is an undesirable artefact that makes the data more difficult to interpret, especially in the presence of static scatterers. Whilst the investigation of instrument response function (IRF) deconvolution and static scatterer compensation techniques could be one way to solve this problem, another solution could be to adopt the approach used by ISVS [19], for example. ISVS works by assessing the amount of *blurring* (or equivalent visibility factor) within recorded holograms, rather than using *frequency shifting*, to access sample dynamics. A longer integration time, or shorter sample decorrelation time, will result in more blurring and vice versa, and Xu et al. derived a quantitative model to relate these three factors, similarly to the approach developed by Hussain et al. [78]. Similar also to FD-DCS, ISVS uses a lensless Fourier holography setup, and therefore only one Fourier transform is required to reconstruct the holographic terms of interest. Additionally, ISVS uses the interferometric nature of holography to compensate for camera noise and access $g_{1}\left(\tau \right)$ directly, thereby obviating the Siegert relationship and the constraints therein. The ISVS system that Xu et al. presented uses a sampling rate of 100 Hz and was able to resolve CBF sample dynamics in vivo using SDS distances of 0.75 and 1.50 cm, using photon count rates that were not high enough to yield a detectable single-mode DCS signal. However, the off-axis setup that was used by these authors was not ideal, as it did not allow for sampling of the reference beam intensity from measured holograms. This lack of calibration leads to noisier flow measurements when compared to like-for-like DCS flow measurements.

The interested reader is referred to [84] for an excellent and fuller treatment of the recent work in the nascent area of interferometric DCS techniques.

### 4.5. Depth Discrimination Techniques

As DCS is a diffuse optical technique, it is limited by a lack of inherent depth discrimination within the illuminated region of each source–detector pair, and the CBF signal is therefore also prone to contamination by the extracerebral tissues which the light traverses. If robustness to extracerebral contamination is the main concern, then the measurement of BFI (i.e., DCS), rather than haemoglobin concentration (i.e., NIRS), will, in theory, provide a more effective signal. This is because optical fluctuations due to BFI achieve 3–5 times better brain specificity than optical absorption, as brain BFI exceeds extracerebral BFI by 6–10 times, while the corresponding ratio for haemoglobin concentration is 2.5 times [6]. Thus an approach to mitigate extracerebral contamination when measuring optical brain signals, and thereby improve depth discrimination, is to measure BFI rather haemoglboin concentration. However, this rationale relies on the SNR of DCS and NIRS being comparable [13]. NIRS, which does not rely on coherence, can measure many incoherent modes using large detectors, and this technique can therefore more easily achieve a higher SNR than DCS. Therefore, changes to improve the SNR of DCS will help to realise this theoretical like-for-like advantage that DCS has over NIRS. DCS has a further advantage over NIRS in that it affords more approaches to brain specificity, including ultrasound modulation and multi-layer models with modulation of probe pressure [70].

One method to account for extracerebral tissue is to switch from the traditional semi-infinite geometry model, which is commonly used in DCS, to a multi-layered (typically two or three-layered) geometry model [85,86,87]. Although this is an effective technique, these models rely on accurate prior knowledge of the optical properties and thickness of each layer. Zhao et al. combined multi-layer modelling with absorption compensation and the zero-lag derivative method (as earlier lags are more sensitive to deeper photon paths) [43]. Alternatively, Selb et al. proposed to remove the contribution of extracerebral contamination via a superficial regression technique, in which a fraction of the BFI of a short SDS pair is subtracted from the BFI of a corresponding long SDS pair [2]. These authors removed a fixed contribution of 70% of the short SDS BFI (i.e., CBF = BFI_long_ − 0.7 × BFI_short_) and noted that this technique could be optimised through further measurements and modelling studies and also with optimisation of the DCS probe location based on observation of a patient’s prior computed tomography (CT) or magnetic resonance imaging (MRI) scans. A natural extension of this technique is the principle of diffuse correlation tomography (DCT), in which a 3D volumetric image of BFI may be reconstructed based on intensity measurements acquired using arrays of sources and detectors [58,88].

Probe pressure modulation is an alternative technique which makes use of a two-layer model (i.e., extracerebral and cerebral layers), with measurements acquired at multiple optical probe pressures and multiple SDS distances, and which does not require prior anatomical information [89]. By exploiting the fact that variations in optical probe pressure will induce variations in extracerebral BFI, whilst cerebral BFI remains constant, these authors were able to isolate the contribution to BFI of the cerebral layer.

Other depth discrimination approaches have also been proposed, including TD-DCS (Section 4.6) and acousto-optic techniques, which make use of the ultrasound tagging of light (UTL), as described in Section 4.7.

### 4.6. Time-Domain Approaches

Aside from contamination by extracerebral tissue, one of the other major limitations of DCS is the need to know the optical properties of tissue to accurately quantify BFI. TD-DCS is a TOF resolved method that enables deep-tissue BFI measurement with depth discrimination at short SDS distances whilst simultaneously acquiring tissue optical properties [12]. In this technique, each detected photon is associated with two measurements: the TOF from source to detector to obtain the temporal point spread function (TPSF) and the absolute arrival time to obtain $g_{2}$. Optical properties can be extracted from the characteristic properties of the TPSF. Then, by using a train of long coherence length laser pulses, autocorrelation functions can be evaluated at different time gates of the TPSF, and flow parameters can then be fit to measured $g_{2}$ functions arising from early and late arriving photons (i.e., short and long photon paths, respectively), which thus provides depth discrimination. Due to the decreased SDS distances that are used, TD-DCS affords both better depth sensitivity and improved spatial resolution than continuous wave DCS (CW-DCS) [59] and as such breaks the trade-off between spatial resolution and cerebral sensitivity.

Compared to CW-DCS, the measurement duty cycle of TD-DCS is decreased by a factor of 50 due to the need for pulsed illumination [12]. Additionally, the intrinsically lower coherence of a pulsed source results in lower SNR. TD-DCS performance is strongly impacted by the system IRF, the selection of time gates that are used, and the temporal shape of the pulsed laser source [90]. Satisfying the TD-DCS requirements of a picosecond pulsed laser with a long coherence length, adequate illumination power, and a narrow IRF remains challenging [91].

TD-DCS at 1064 nm has recently been described by both Ozana et al. [69] and Poon et al. [92] which required the use of a pulse-shaped long-coherence laser at 1064 nm. Renna et al. are currently researching *multispeckle* TD-DCS at 1064 nm, which was initially presented by way of 4 SNSPD units operating in parallel [93], but plans are in place to progress toward a fast-gated 32×32 InGaAs/InP SPAD array that makes use of multichannel FPGA based autocorrelation for up to 192 channels [62,67,94].

An alternative technique, interferometric near-infrared spectroscopy (iNIRS), also quantifies TOF resolved dynamics, and this is introduced in Section 5.

### 4.7. Other Approaches

Robinson et al. have presented an approach toward enabling quantitative depth selective flow measurements in DCS by using acousto-optic modulation, namely AOM-DCS [20]. This is a hybrid technique that allows for improved spatial resolution of the optical signal based on knowledge of the area that is insonified by ultrasound, owing to the UTL effect. These authors presented a quantitative model for flow detection using continuous wave ultrasound based on the principle that both $g_{2}\left(\tau \right)$ and the amplitude of the modulation of $g_{2}\left(\tau \right)$ decay with time [95]. More complex spatiotemporal distributions of ultrasound (e.g., focused, pulsed, encoded, or overlapping pressure fields using two ultrasound transducers) are required to resolve flow with both better spatial and temporal resolution. AOM-DCS is part of a broader group of methods, known as AOT, a full discussion of which is outside the scope of this paper but is available in [40,51,96]. The interaction of the ultrasound generated acoustic radiation force (ARF) and the DCS signal has also been investigated [97].

With a view to developing a low-cost and wearable DCS system, Biswas et al. have recently presented a portable DCS system which makes use of a small-form factor and fibre-less embedded diode laser [98]. The same group has also developed a system to obtain pathlength resolved DCS measurements (similar to TD-DCS and iNIRS) by using a Mach–Zehnder interferometer with a reference arm that has an adjustable length [99]. By scanning the length of the reference arm, $g_{2}\left(\tau \right)$ may be evaluated for various photon pathlengths. Additionally, the average intensity as a function of pathlength can be used to generate a TPSF, from which tissue optical properties may be obtained.

Deep learning techniques have recently been employed with a view to reducing the computational demand of DCS experiments [100]. Fitting measured $g_{2}\left(\tau \right)$ data to analytical or Monte Carlo models to extract BFI can be computationally demanding and suffers from inaccuracy in low SNR environments. These authors used a deep learning model that resulted in a 23-fold increase in the speed of BF quantification, which further enables the real-time and accurate quantification of BFI using DCS. DCS denoising algorithms, including the use of support vector regression, have also been explored [101]. Within the context of DCT image reconstruction, learning approaches have also been employed to compensate for the decreased spatial resolution resulting from the ill-conditioned and ill-posed nature of the inverse problem, resulting in an imaging depth of 5 mm at 2.5 Hz using a 32 × 32 SPAD array [102].

## 5. Interferometric Near-Infrared Spectroscopy

Frequency-domain NIRS (FD-NIRS) and DCS instruments can be combined into one instrument to concurrently measure both optical properties and sample dynamics; however, this technique cannot provide TOF resolved DCS measurements. To overcome this limitation, iNIRS combines time-domain NIRS (TD-NIRS) and DCS into a single modality to simultaneously extract optical properties and TOF resolved sample dynamics (Figure 3). This is achieved through the analysis of a spectral interference fringe pattern, which is measured using a Mach–Zehnder interferometer with a frequency-swept narrow linewidth laser [103]. Light with a shorter pathlength will produce a lower frequency fringe pattern, and arrive at the detector earlier than light with a longer pathlength, which will produce a higher frequency fringe pattern.

The TOF difference between light propagating through the sample arm and the reference arm of the interferometer can be obtained by Fourier transforming the frequency resolved interference signal. This yields a complex mutual coherence function, $\Gamma_{rs}\left(\tau_{s},t_{d}\right)$, between the sample and reference fields, where $\tau_{s}$ is TOF and $t_{d}$ is delay time. From a time series in $t_{d}$, a 2D TOF resolved optical field autocorrelation function, $G_{1}^{\mathrm{iNIRS}}\left(\tau_{s},\tau_{d}\right)$, can be obtained: (16)$$G_{1}^{\mathrm{iNIRS}}\left(\tau_{s},\tau_{d}\right)={\langle \Gamma_{rs}^{*}\left(\tau_{s},t_{d}\right)\Gamma_{rs}\left(\tau_{s},t_{d}+\tau_{d}\right)\rangle }_{t_{d}},$$
where $\tau_{d}$ is time lag. The measured $G_{1}^{\mathrm{iNIRS}}\left(\tau_{s},\tau_{d}\right)$ is related to the intrinsic $G_{1}\left(\tau_{s},\tau_{d}\right)$ by convolution with the system IRF,
(17)$$G_{1}^{\mathrm{iNIRS}}\left(\tau_{s},\tau_{d}\right)=G_{1}\left(\tau_{s},\tau_{d}\right)*\mathrm{IRF}\left(\tau_{s}\right),$$
where ∗ denotes convolution. $G_{1}^{\mathrm{iNIRS}}\left(\tau_{s},0\right)$ is equivalent to the TPSF that is measured from TD-NIRS. $G_{1}^{\mathrm{iNIRS}}\left(\tau_{s},\tau_{d}\right)$ provides TOF resolved dynamics, which, when integrated over $\tau_{s}$ and normalised, yields $g_{1}\left(\tau_{d}\right)$, as per a conventional DCS experiment.

Therefore, $G_{1}^{\mathrm{iNIRS}}\left(\tau_{s},\tau_{d}\right)$ is a rich data set which provides both sample optical properties and TOF resolved sample dynamics, and the authors of this technique argue that it is the most informative diffuse optical method to date. Advantages of iNIRS, when compared to TD-DCS, for example, are that iNIRS does not require pulsed lasers (which result in a reduced coherence factor) or single photon counting detectors, and TD-DCS is also sensitive to ambient light. iNIRS also accesses the field autocorrelation directly, therefore obviating the Siegert relation and the constraints therein.

The main drawback of iNIRS, when compared to incoherent TD-NIRS for example, is that iNIRS is a coherent modality which, like traditional DCS techniques, suffers from moderate light throughput. In order to compensate for this lack of parallelisation, continuous-wave parallel iNIRS (CW πNIRS) has recently been published [105]. This technique uses a fast 2D CMOS sensor operating at 1.1 MHz for 2048 pixels (which has a higher active area than a SPAD array). The authors typically used this instrument at 0.6 MHz to increase the number of available pixels to 8192 whilst maintaining a minimum lag time of 1.67 μs. Using ∼8000 parallel channels allowed the authors to sense BF with an SDS distance of 2–3 cm using integration times as short as 2–10 ms, which is 50–100 times faster than single-channel DCS techniques. However, owing to the data rate involved with this technique, measurements could only last for several seconds due to the limited internal memory of the camera. Furthermore, although the frame rate of 1.1 MHz is sufficient to capture sample dynamics due to speckle decorrelation, it is still too slow to resolve TOF resolved dynamics when employing a swept laser source. Achieving TOF resolution using πNIRS would require at least a two-order-of-magnitude increase in the speed of cameras that are currently available. A further limitation of this technique is the current high cost of the detector (∼USD 50,000).

## 6. Summary and Outlook

In this paper, we have developed a description of the currently vacant niche for an inexpensive, continuous, noninvasive, portable, bedside, and non-ionising imaging/sensing modality with which to measure CBF, to which biomedical optics solutions are well suited. Under this umbrella of techniques, diffuse optics is required to image deeper into the brain; however, these techniques are marred by the low SNR that is associated with the SDS distances that are required to achieve deeper imaging. DCS is a promising CBF measurement technique, and many research groups have recently been investigating methods to improve the SNR, imaging depth, and spatial resolution of DCS. Such methods have included multispeckle, long-wavelength, interferometric, and depth discrimination techniques. TD-DCS and iNIRS are perhaps the most comprehensive diffuse optical imaging modalities to date and, with recent advancements in parallelisation strategies, may be the most applicable to the measurement of CBF. In this regard, a standardised performance assessment of DCS instruments is necessary in order to benchmark competing candidate technologies [106].

The ultimate goal of DCS is to acquire both high photon throughput and depth discrimination concurrently, but this goal is yet to be accomplished. Promising routes to achieve this objective are long wavelength, parallelised, interferometric, and TOF resolved DCS technology, which is compatible with future developments in AOT and which makes use of inexpensive detectors and a data rate that is feasible for real time processing. With the advances that have been made in DCS since 2016, we expect exciting developments which fulfil these criteria in the years to come.

## Figures and Tables

**Figure 1 sensors-23-09338-f001:**
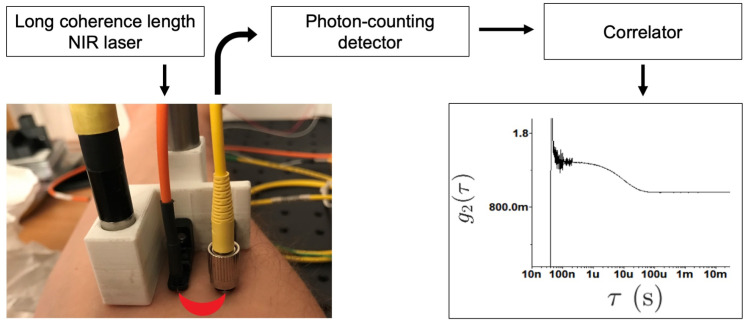
Schematic overview of a diffuse correlation spectroscopy (DCS) system and measurement pipeline. Source and detector fibres are mounted on the sample surface. Artefacts are present at very short delay times in $g_{2}\left(\tau \right)$, arising from downsampling due to hardware constraints. As such, $g_{2}\left(\tau \right)$ is thresholded at a minimum lag time prior to further data processing. Reproduced with permission from [32] under the terms of a CC BY-NC 4.0 License. © James 2022.

**Figure 2 sensors-23-09338-f002:**
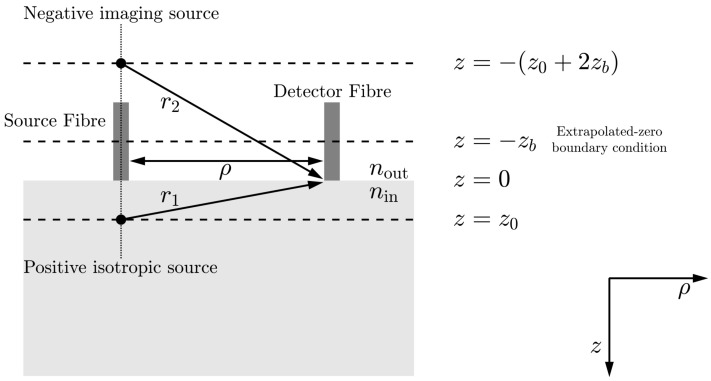
Semi-infinite geometry model and notation used in DCS measurements. The grey area is the medium to be examined, the white area is the surrounding air, and the interface is at z=0. The collimated source is approximated as a positive isotropic imaging source located at $z=z_{0}$. The extrapolated zero-boundary condition requires a signal size of zero at $z=-z_{b}$ [49], which can be achieved by modelling a negative isotropic imaging source located at $z=-(z_{0}+2z_{b})$. ρ represents the source-detector separation (SDS) distance. Adapted from [21]. Reproduced with permission from [50] under the terms of a CC BY 4.0 License. © James and Powell 2020.

**Figure 3 sensors-23-09338-f003:**
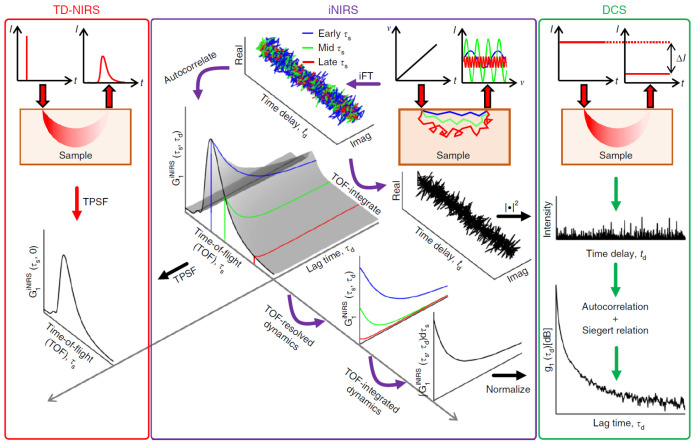
Due to the fact that the laser frequency is swept, light travelling deeper into tissue, which takes a longer time to travel through the sample arm of the interferometer, generates a larger beat frequency upon interference with the reference arm. Therefore, interferometric near-infrared spectroscopy (iNIRS) encodes time of flight (TOF), $\tau_{s}$, as beat frequency. Autocorrelation of a time-series of mutual coherence functions yields $G_{1}^{\mathrm{iNIRS}}\left(\tau_{s},\tau_{d}\right)$, which can be used to obtain both temporal point spread functions (TPSFs) and TOF resolved dynamics. Reproduced with permission from [104] under the terms of a CC BY 4.0 License. © Kholiquov, Zhou, Zhang, Du Le, and Srinivasan 2020.

## Data Availability

No new data were created or analysed in this study.

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
