# Peer review of "Diffuse Correlation Spectroscopy: A Review of Recent Advances in Parallelisation and Depth Discrimination Techniques"

_sensors, 2023, doi:10.3390/s23239338_

Round 1

Reviewer 1 Report

Comments and Suggestions for Authors

I have evaluated your manuscript, titled "Diffuse correlation spectroscopy: a review of recent advances in parallelization and depth discrimination techniques," and I am confident that it is well-suited for publication in our journal.

The article is elegantly written, and the inclusion of mathematical formulas enhances readers' comprehension of the technology. Through this comprehensive review, readers can gain a profound understanding of the principles, applications, and the latest advancements in DCS technology. Your article provides valuable insights for researchers and scholars in this field, and it is poised to positively influence the future development of DCS technology. I eagerly anticipate the submission of your final revised version, and once more, I express my gratitude for your significant contribution to this field.

Author Response

Reviewer 1

I have evaluated your manuscript, titled "Diffuse correlation spectroscopy: a review of recent advances in parallelization and depth discrimination techniques," and I am confident that it is well-suited for publication in our journal.

The article is elegantly written, and the inclusion of mathematical formulas enhances readers' comprehension of the technology. Through this comprehensive review, readers can gain a profound understanding of the principles, applications, and the latest advancements in DCS technology. Your article provides valuable insights for researchers and scholars in this field, and it is poised to positively influence the future development of DCS technology. I eagerly anticipate the submission of your final revised version, and once more, I express my gratitude for your significant contribution to this field.

We thank the reviewer for the appreciation of this work.

Reviewer 2 Report

Comments and Suggestions for Authors

The subject of this review of diffuse correlation spectroscopy (DCS). It is focused on recent advances in parallelisation and depth discrimination techniques. There are almost 100 references in this review, most of them are dated after 2015. There are 3 figures with relevant clarifications and copyright references. The review is structured into 6 sections.

The introduction contains a concise overview of the early states of the DCS  development in 1997 and shortly after. In Sections 2 and 3, the motivation and algorithms behind DCS are provides. A geometry model and notations used in DCS are discussed in detail in Section 4. Key limitations of DCS are considered in Subsection 4.1. Moreover, in Section 5, interferometric near-infrared spectroscopy on pages 14-16. The most topical part of the review is placed in Subsections 4.2-4.6, in which Interferometric, long wavelength, multispeckle, depth discrimination and time-domain approaches to improve and overcome the limitations of the basic DCS approach are reviewed and discussed. A short summary and outlook are given in Section 6.  

This review is scientifically sound and interesting for beginners and experts in diffuse correlation spectroscopy, especially Subsections 4.2-4.6 focused on recent advances and algorithms to improve DCS. English is clear and easy to understand. This review is a valuable contribution to the field, it can be recommended to Sensors as it is. 

Author Response

Reviewer 2

The subject of this review of diffuse correlation spectroscopy (DCS). It is focused on recent advances in parallelisation and depth discrimination techniques. There are almost 100 references in this review, most of them are dated after 2015. There are 3 figures with relevant clarifications and copyright references. The review is structured into 6 sections.

The introduction contains a concise overview of the early states of the DCS  development in 1997 and shortly after. In Sections 2 and 3, the motivation and algorithms behind DCS are provides. A geometry model and notations used in DCS are discussed in detail in Section 4. Key limitations of DCS are considered in Subsection 4.1. Moreover, in Section 5, interferometric near-infrared spectroscopy on pages 14-16. The most topical part of the review is placed in Subsections 4.2-4.6, in which Interferometric, long wavelength, multispeckle, depth discrimination and time-domain approaches to improve and overcome the limitations of the basic DCS approach are reviewed and discussed. A short summary and outlook are given in Section 6.  

This review is scientifically sound and interesting for beginners and experts in diffuse correlation spectroscopy, especially Subsections 4.2-4.6 focused on recent advances and algorithms to improve DCS. English is clear and easy to understand. This review is a valuable contribution to the field, it can be recommended to Sensors as it is. 

We thank the reviewer for the appreciation of this work.

Reviewer 3 Report

Comments and Suggestions for Authors

Author Response

Reviewer 3

This review is very well written. It gives adequate and appropriate background information for any reader to learn about this topic. The review has comprehensive coverage of related work. The authors arranged the sections reasonably. I especially appreciate that besides including all the related work, the authors elaborated the rationale and key features of each work clearly. I have the following comments that may hopefully further improve the quality of paper.

We thank the reviewer for the appreciation of this work.

Major comment:

  1. The authors reviewed iDWS with variable TOF filter in the iNIRS section. However, in principle, this work is more related to iDWS rather than iNIRS. I suggest the authors to move the review of this work to either 4.4 or 4.6. please see this talk for reference: https://doi.org/10.1117/12.2649073 (towards the end, a parametric comparison of interferometric diffuse optics techniques).

We thank the reviewer for highlighting this interesting conference presentation. We agree that the review of iDWS with a variable TOF filter is more related to iDWS than iNIRS, and have moved the review of this work to Section 4.4. We have also included a reference to this conference presentation.

Minor comments:

  1. Line 282, DCS dependence on mu_a and mu_s’: I suggest to cite this paper: D. Irwin, L. Dong, Y. Shang, R. Cheng, M. Kudrimoti, S. D. Stevens, G. Yu, “Influences of tissue absorption and scattering on diffuse correlation spectroscopy blood flow measurements,” Biomedical Optics Express, 2, 1969– 1985 (2011)

We thank the reviewer for pointing out this reference. We have read through the suggested reference and agree that it is helpful to include it here as a reference here, and have done so.

  1. Line 438, ~96 speckles: I would suggest the authors to give “effective independent channel count” (= 2*speckle number) instead of speckle count, which is easier for readers to compare to “channel count” of conventional DCS.

We disagree with the reviewer on this point. It is generally accepted in the literature that the effective independent channel count is equal to the speckle number. This is demonstrated in detail in reference 42, as well as reference 16, which specifically addresses interferometric methods. No changes have been made here.

  1. Line 703: I personally think the ultimate goal is high throughput and depth discrimination. “Long wavelength, parallelization, interferometric, TOF resolved” are the promising approaches to the ultimate goal.

We agree that the ultimate goal of DCS could be made more concise here, and we have reworded the final paragraph to take this into consideration.

Reviewer 4 Report

Comments and Suggestions for Authors

This is an interesting 2023 review work for the advances in Diffuse Correlation Spectroscopy (DCS), a non-invasive optical modality for indicating blood flow in real-time. The overall structure and language are very good. However, there are essential points that need to be addressed.

MAJOR POINTS

1] Line 13: is the DCS technique only applicable to the brain tissue? It seems that it could be used for every tissue in the human body.

2] Line 66: How do we measure Intracranial Pressure (ICP)? Is there a way to measure ICP noninvasively?

3] Line 74-75: According to Ohm’s law, CBF = (MAP – VP) / CVR, where VP is Venous Pressure. ICP affects both MAP and VP inside the skull and when ICP becomes greater than VP, it starts to affect CBF.

4] Lines 158-162: The physical meaning of the principal measured quantities of V2 and Db should be further analyzed. Lines 224-232 should be incorporated here, underscoring the importance of the microvasculature. If the Brownian model fits observed data more precisely (Line 227), why do we still use the convective flow model?

5] Line 181: it should be explained why a long coherence length is required.

6] Lines 189-190: it is stated that “…the shape and the rate of decay of g2(t) provides information corresponding to the nature and the speed of the motion of scatterers…”, however, it is not immediately apparent how exactly this is done from the experimental curve of Figure 1, and taking into account Lines 219-221.  A] An example of estimating V2 and Db from an experimental curve would be helpful.  B] Under what condition is it possible to assume a linear combination of a Brownian and a convective motion model (Line 220)?             C] Were there any efforts to in-vitro calibrate the estimation of V2 and Db?

7] Line 205: μα and μs were not defined. It should also be described how they are measured since DCS depends on their measurement (Line 282).

ADDITIONAL POINTS

8] Line 11: Since this is a review article, there is an opportunity to add an introductory paragraph referring to other related spectroscopic techniques and research, such as Near-InfraRed Spectroscopy NIRS (Nogueira Soares et al 2020, Koutsiaris AG 2017, Harrison DK et al 1998), Raman spectroscopy (Butler et al 2016), and Infrared spectroscopy (Putzig CL et al 1994).

9] Line 15: the “noninvasive brain-computer interface” is a formidable potential application which is mentioned again in lines 287-288 so, it would be useful to the broader research community to briefly describe how DCS is involved in it.

10] Line 30: instead of the term “short-wave infrared” it would be better to use the well-established term “near-infrared (NIR)”. More accurate terms would be “NIR-CW-DCS” and “NIR-TD-DCS” since all DCS systems operate in the (650 - 1300 nm) NIR optical window. It would be nice to refer to the pioneering work of F Jobsis (1977, Science).

11] Line 103: The authors miss mentioning the other imaging modalities that attempt to measure CBF.

12] Line 206: Κο was not defined.

13] Line 248: “single-mode” should be explained.

14] Line 669: the word “separation” should change to another one.

Author Response

Reviewer 4

This is an interesting 2023 review work for the advances in Diffuse Correlation Spectroscopy (DCS), a non-invasive optical modality for indicating blood flow in real-time. The overall structure and language are very good. However, there are essential points that need to be addressed.

We thank the reviewer for the appreciation of this work.

MAJOR POINTS

1] Line 13: is the DCS technique only applicable to the brain tissue? It seems that it could be used for every tissue in the human body.

DCS does have applications in many areas of the body, although the vast majority of the DCS literature focuses on CBF measurement. We agree that it would be useful to introduce the reader to other applications of DCS, and have added text and references toward the end of Section 2 outlining the application of DCS to monitoring cancer, peripheral arterial disease, muscle disease, exercise physiology, and myocardial infarction.

2] Line 66: How do we measure Intracranial Pressure (ICP)? Is there a way to measure ICP noninvasively?

A review of ICP monitoring techniques, although a very interesting topic is out of scope for this review paper, and deserves a dedicated review paper. In this review we have discriminated between invasive and non-invasive ICP measurement, discussed that in certain populations of patients clinicians would prefer to avoid invasive ICP monitoring, and sign-posted the reader to reference 31 for a recent review of non-invasive ICP monitoring methods. We have amended the sentence “clinicians would prefer to avoid the risks of invasive continuous cerebral monitoring devices” to “clinicians would prefer to avoid the risks of invasive ICP measurement using continuous cerebral monitoring devices”.

3] Line 74-75: According to Ohm’s law, CBF = (MAP – VP) / CVR, where VP is Venous Pressure. ICP affects both MAP and VP inside the skull and when ICP becomes greater than VP, it starts to affect CBF.

The normal range of ICP is 8-13 mmHg and the normal range of venous pressure outside the dura is 0-5 mmHg, and therefore we’re not sure how relevant it is to include VP in the above equations. It is true that ICP should be greater than VP, and that this affects CBF (which is demonstrated by Equations 1 and 3 of the paper).

We have updated the following sentence “CA is the intrinsic homeostatic mechanism by which CBF is kept within tightly controlled bounds, in the face of variations in mean arterial pressure (MAP) and intracranial pressure (ICP) [23], which itself can affect arterial and venous pressure changes within the skull.”

4] Lines 158-162: The physical meaning of the principal measured quantities of V2 and Db should be further analyzed. Lines 224-232 should be incorporated here, underscoring the importance of the microvasculature. If the Brownian model fits observed data more precisely (Line 227), why do we still use the convective flow model?

When the physical meaning of the measured quantities is first introduced, we discuss how V2 is related to convective scatterer motion, and how Db­­ is related to Brownian motion. We also discuss how other types of particle motion are related to the measured signal.

We later emphasise the importance of microvasculature on the measured signal. We appreciate the reviewer’s point here and have moved this section up to the preceding paragraph.

The Brownian model fits the data better than a convective model, but not better than a mixed model that combines the two. That is why we still use a convective model. We have therefore added to the end of this section – “Previous authors have found that the Brownian model fits observed data more precisely than the convective flow model in many biological tissues [21], although a mixed model can also be used [47], as shown by Equation 11.”

5] Line 181: it should be explained why a long coherence length is required.

We agree that some further explanation would be helpful here and have added - “A long coherence length is used to provide the necessary conditions for interference over the diffuse optical pathway between the source and the detector.”

6] Lines 189-190: it is stated that “…the shape and the rate of decay of g2(t) provides information corresponding to the nature and the speed of the motion of scatterers…”, however, it is not immediately apparent how exactly this is done from the experimental curve of Figure 1, and taking into account Lines 219-221.  A] An example of estimating V2 and Db from an experimental curve would be helpful.  B] Under what condition is it possible to assume a linear combination of a Brownian and a convective motion model (Line 220)?             C] Were there any efforts to in-vitro calibrate the estimation of V2 and Db?

We have added a signpost for the interested reader to see an example of the model fitting process; however, we would rather focus on the review of depth discrimination and parallelisation techniques, which is the main focus of this paper. This signpost also contains examples on mixed model fitting, and the calibration of Db values in an intralipid phantom.

We have added the following to the text “The interested reader is referred to [50] for examples of fitting Db and V2 to experimental data, and also for examples of mixed model fitting and the validation of measured Db values in an intralipid phantom.”

The condition for assuming the linear model is that red blood cells are the only source of dynamic scattering events, and we have amended the text as follows - “by fitting to either a Brownian or a convective motion model of Equation 10, or a linear combination of the two, assuming that RBCs are the only source of dynamic scattering events and that [47] ….”

7] Line 205: μα and μs were not defined. It should also be described how they are measured since DCS depends on their measurement (Line 282).

Thank you for pointing this out, we have now defined \mu_a and \mu_s' (optical properties) directly after their first use. We have already mentioned at the beginning of section 4.6 that one of the major limitations of DCS is the need to know the optical properties of tissue to accurately quantify BFI, and in response to comment by reviewer 3 we have included a further reference on this (reference 57). We describe at the beginning of Section 5 how NIRS and DCS instruments can be combined to measure both optical properties and sample dynamics. Throughout the paper we discuss novel DCS techniques that allow for both the measurement of optical properties and sample dynamics (e.g. TD-DCS and iNIRS).

ADDITIONAL POINTS

8] Line 11: Since this is a review article, there is an opportunity to add an introductory paragraph referring to other related spectroscopic techniques and research, such as Near-InfraRed Spectroscopy NIRS (Nogueira Soares et al 2020, Koutsiaris AG 2017, Harrison DK et al 1998), Raman spectroscopy (Butler et al 2016), and Infrared spectroscopy (Putzig CL et al 1994).

We understand that NIRS and DCS are highly complementary technologies in diffuse biomedical optics. As such we have interleaved references to NIRS where relevant in this DCS paper. These references start at the end of Section 4.0, where we discuss DCS and NIRS measurements. In section 4.3, we discuss how long wavelength approaches, whilst appropriate for DCS, are not appropriate for NIRS. We discuss the fundamental SNR difference between NIRS and DCS for CBF measurement in Section 4.5. Finally, we discuss frequency domain NIRS and time domain NIRS in Section 5, and how these two relate to iNIRS.

We do not believe that an introduction to Raman or Infrared Spectroscopy would add to the paper.

9] Line 15: the “noninvasive brain-computer interface” is a formidable potential application which is mentioned again in lines 287-288 so, it would be useful to the broader research community to briefly describe how DCS is involved in it.

We appreciate that some more information would be helpful here and have updated lines 100-104 to read “Finally, CBF monitoring also has applications in the development of a noninvasive brain-computer interface [3,4], although these authors have demonstrated that sensitivity to local, subsecond latency haemodynamic response of the cerebral cortex at a depth of 15 mm below the scalp surface is not feasible with current detector technology and signal processing techniques.”

10] Line 30: instead of the term “short-wave infrared” it would be better to use the well-established term “near-infrared (NIR)”. More accurate terms would be “NIR-CW-DCS” and “NIR-TD-DCS” since all DCS systems operate in the (650 - 1300 nm) NIR optical window. It would be nice to refer to the pioneering work of F Jobsis (1977, Science).

We appreciate the point that the reviewer makes here, indeed all DCS techniques operate in the NIR window, as the reviewer has pointed out. DCS was traditionally performed at 785 nm, and as described in Section 4.3 of this paper, recent efforts have shown that moving to a longer wavelength of 1064 nm will confer SNR benefits. The authors of this shift to longer a longer wavelength refer to this wavelength as SWIR (references 13-15), which is 900 nm to 1,700 nm, and for this reason we would prefer to be consistent with this.

11] Line 103: The authors miss mentioning the other imaging modalities that attempt to measure CBF.

It is true that there are many other imaging modalities to measure CBF, and regretfully, we must leave a full review of these as the topic of a future review paper. We have added a sentence here to signpost the reader to reviews of CBF measurement methods (including both non-optical and optical methods), and have already mentioned that many modalities exist, each of which has its own set of advantages and disadvantages, and that no ideal modality exists.

12] Line 206: Κο was not defined.

We have defined wavenumber here after its first use, thank you for pointing this out.

13] Line 248: “single-mode” should be explained.

Single-mode has already been introduced and explained at the beginning of Section 3, with a signpost to reference 42. The term is then explored further at the beginning of Section 4, before its use at the beginning of Section 4.1 as described here. We believe that any further explanation of the term single-mode would be unnecessary at this point.

14] Line 669: the word “separation” should change to another one.

Thank you for pointing this out, we have changed this to distance.

Round 2

Reviewer 3 Report

Comments and Suggestions for Authors

The authors addressed most of the comments nicely. I still have reservations about the definition of "independent channel" for interferometric DCS/DWS systems. But it is clear that "~96 speckles" is accurate and avoids any ambiguity/argument. So I agree with the authors to report "~96 speckles". 

Author Response

This reviewer recommends publication. 

Reviewer 4 Report

Comments and Suggestions for Authors

This REVIEW paper is now at an appropriate level for publication.

Author Response

This reviewer recommends publication.